# Current Treatments for Diabetic Macular Edema

**DOI:** 10.3390/ijms24119591

**Published:** 2023-05-31

**Authors:** Tomoaki Tatsumi

**Affiliations:** Department of Ophthalmology and Visual Science, Chiba University Graduate School of Medicine, Inohana 1-8-1, Chuo-ku, Chiba 260-8670, Japan; ttatsumi@chiba-u.jp; Tel.: +81-43-226-2124; Fax: +81-43-224-4162

**Keywords:** diabetic macular edema, retinal neurovascular unit, diabetic neuropathy, diabetic retinopathy, anti-vascular endothelial growth factor

## Abstract

Diabetic retinopathy is a major retinal disorder and a leading cause of blindness. Diabetic macular edema (DME) is an ocular complication in patients with diabetes, and it can impair vision significantly. DME is a disorder of the neurovascular system, and it causes obstructions of the retinal capillaries, damage of the blood vessels, and hyperpermeability due to the expression and action of vascular endothelial growth factor (VEGF). These changes result in hemorrhages and leakages of the serous components of blood that result in failures of the neurovascular units (NVUs). Persistent edema of the retina around the macula causes damage to the neural cells that constitute the NVUs resulting in diabetic neuropathy of the retina and a reduction in vision quality. The macular edema and NVU disorders can be monitored by optical coherence tomography (OCT). Neuronal cell death and axonal degeneration are irreversible, and their development can result in permanent visual loss. Treating the edema before these changes are detected in the OCT images is necessary for neuroprotection and maintenance of good vision. This review describes the effective treatments for the macular edema that are therefore neuroprotective.

## 1. Introduction

The global prevalence of diabetes in 20- to 79-year-olds was estimated to be 10.5% (536.6 million) in 2021, and it is predicted to rise to 12.2% (783.2 million) in 2045 [1]. Diabetic retinopathy, a major complication of diabetes, is one of the leading causes of vision reduction [2]. The vision threatening stages of diabetic retinopathy are proliferative diabetic retinopathy with tractional retinal detachments, vitreous hemorrhages, neovascular glaucoma and diabetic macular edema (DME). According to a meta-analysis, 10.2% of the 22,896 individuals with diabetes had diabetic retinopathy, and 6.81% had DME [3]. DME is the most common retinal disorder other than proliferative diabetic retinopathy that can lead to vision reduction in patients (PDR) [4]. In recent years, intravitreal injections of anti-vascular endothelial growth factor (anti-VEGF) agents are recognized as the most effective treatment for DME. This treatment has achieved the status of the globally accepted treatment for DME [5,6]. However, there is a high rate of recurrence, and frequent injections are required to prevent vision loss [7]. It has been reported that DME persisted in 31.6 to 65.6% of the patients, even after multiple anti-VEGF injections [8].

## 2. Pathogenesis of Diabetic Retinopathy and DME

All of the retinal cells are interdependent, and they interact to maintain a healthy environment and function appropriately. It has been proposed that the retinal dysfunction associated with diabetes may be due to changes in the retinal neurovascular units (NVUs) [9]. The NVUs are composed of neural cells (ganglion cells, amacrine cells, horizontal cells, and bipolar cells), glial cells (Müller cells, microglia, and astrocytes), and vascular cells (endothelial cells, pericytes, and basement membrane). The interactions of the glial cells, pericytes, and neurons promote the formation and maintenance of the blood–retina barriers which control the flux of fluids and bloodborne metabolites into the neural parenchyma [9].

Several studies have reported that neuronal abnormalities, including neuronal cell death, occur prior to the clinical findings associated with vascular abnormalities, although this has not been determined conclusively. This suggestion was based on the findings in animal models [10,11,12,13,14], in human retinas [15,16,17], in the clinical optical coherence tomographic (OCT) findings [18,19,20], and in the electroretinographic (ERG) findings [21,22,23].

Many biomarkers of DME have been found in the OCT images, and they are very helpful for the evaluations and treatments of DME [24]. The OCT images of the outer retina provide useful information about the photoreceptors. The status of the external limiting membrane (ELM), whether it is intact or disrupted, is significantly correlated with the visual acuity [25]. The integrity of the ELM and the inner and outer photoreceptor segment junctions, viz., the ellipsoid zone (EZ), was more strongly correlated with the best-corrected visual acuity (BCVA) than the central subfield thickness in eyes with DME [26]. Cases with a completely continuous EZ have been shown to have better visual outcomes after treatment for the macular edema than those with disrupted EZs [27]. The disruption of the EZ is correlated with a significant decrease in the focal sensitivities in eyes with DME [28], and it is an important preoperative predictor of the visual acuity in DME patients [29]. The integrities of the EZ and the ELM are useful markers that can be used to evaluate the normality of the foveal photoreceptor layer, and they are closely associated with the final (after treatment) visual acuity in eyes with DME [30]. There has been a study that reported that the integrity of the EZ improved after treatments by anti-VEGF agents [31].

The EZ in the OCT images represents the clustering of mitochondria in the inner segments of the photoreceptors and is the junction between the inner and outer photoreceptor segments. The ELM is present between the photoreceptor layer and the outer nuclear layer that contains the nuclei of the cone and rod photoreceptors [32]. A disruption of the EZ or the ELM represents a neuronal impairment of the retina.

The vascular system and the nervous system are closely related in the NVU, and when edema develops due to an angiopathy, the neuropathy worsens. If the edema persists, there is a progression to irreversible neuropathy. Suppressing angiopathy and hyperpermeability around macula and maintaining an edema-free environment leads to a normalization of the neurovascular units and neuroprotection. In addition, treatment of the macular edema improves the disrupted EZ [31] and neuronal impairments, and it leads to a regeneration of the outer segments of the photoreceptor layer.

## 3. Treatments for Diabetic Macular Edema

### 3.1. Anti-VEGF Treatment

Currently, anti-VEGF treatments are established as first-line therapy for DME, and both preclinical and clinical research have verified the significance of VEGF as a key mediator in DME and proliferative diabetic retinopathy. Thus, anti-VEGF agents were developed to treat ocular diseases, and a large-scale clinical trial showed their positive effects on DME. There are five anti-VEGF agents currently in use for DME in Japan, viz., ranibizumab, aflibercept, bevacizumab, faricimab, and brolucizumab. The molecular characteristics and general clinical information for these anti-VEGF agents are shown in Table 1 [33,34,35], and Figure 1 shows the molecular structure of these anti-VEGF agents. It is important to be aware that some agents cannot be used or are rarely used in some countries or regions, and agents other than those mentioned here can be used. In addition, new agents that appear in the future may change the situation completely.

#### 3.1.1. Ranibizumab

Ranibizumab is the fragment antigen binding (Fab) of a humanized monoclonal antibody against VEGF, and it inhibits VEGF-A that plays an important role in neovascularization and vascular leakage [36]. Ranibizumab is only the Fab region of the full-length immunoglobulin (IgG) (humanized anti-VEGF antibody, molecular weight: 150 × 10^3^), so it has a low molecular weight of 48 × 10^3^. The low molecular weight makes it more diffusible than the full-length IgG. The penetration of the Fabs into the retina is rapid and complete, whereas most of the IgGs are stopped either by the internal limiting membrane (ILM) or by the outer plexiform layer [37].

The clinical efficacy of ranibizumab for DME was shown in the RISE and RIDE studies. These studies were phase III, two parallel, methodologically identical, double-masked, multicenter, sham injection-controlled, randomized trial of ranibizumab monotherapy with sham injections for DME [38]. A total of 759 patients were enrolled and randomized to study the effects of the treatments (377 patients in the RISE and 382 in the RIDE studies). The mean BCVAs improved from the baseline at 24 months in the RISE study: 12.5 letters in the 0.3 mg ranibizumab group and 11.9 letters in the 0.5 mg ranibizumab group. The mean BCVAs gain at 24 months from the baseline in the RIDE study was 10.9 letters in the 0.3 mg ranibizumab group and 12.0 letters in the 0.5 mg ranibizumab group.

In the RISE study, 44.8% of patients receiving 0.3 mg ranibizumab and 39.2% of patients receiving 0.5 mg ranibizumab gained ≧15 letters compared with 18.1% of sham-treated patients. In the RIDE study, the corresponding changes were 33.6%, 45.7%, and 12.3%, respectively.

Intravitreal injections of bevacizumab and aflibercept reduced the plasma VEGF significantly for up to 4 weeks, whereas ranibizumab led to no such effects [39]. Adverse events related to the plasma VEGF levels, such as cardiovascular and cerebrovascular events, did not differ significantly in the clinical trials [40]. However, this suggested that the incidence of systemic adverse events with ranibizumab may not be as high as with the other anti-VEGF agents.

#### 3.1.2. Aflibercept

Aflibercept is a recombinant protein (molecular weight: 115 × 10^3^*)* composed of the fragment crystallizable (Fc) of human IgG1 and the second and third extracellular domains of human VEGF recepotor-1 (VEGFR-1) and VEGF receptor-2 (VEGFR-2) [35]. The aflibercept molecules competitively bind to all VEGF-A and VEGF-B isoforms with higher affinities than their native receptors. Aflibercept can also bind to placental growth factor-1 (PlGF-1) and placental growth factor-2 (PlGF-2), which are other members of the VEGF family that have the potential to induce retinal neovascular changes [35]. A comparison of the equilibrium dissociation constants of anti-VEGF agents aflibercept, ranibizumab, and bevacizumab to VEGF-A_165_ as kinetic binding affinity parameters are 0.49, 46, and 58, respectively [41]. Aflibercept has been shown to bind most strongly to VEGF-A_165_.

The VIVID and VISTA trials compared 2 mg of intravitreal aflibercept every 4 or 8 weeks to macular laser photocoagulation for DME [5]. The mean BCVAs improved from the baseline at 100 weeks in the VISTA group of eyes treated by aflibercept every 4 weeks. There was an improvement of 11.5 letters in the group that was treated with aflibercept every 8 weeks and an improvement of 6.3 letters in the laser group (*p* = 0.0002 and *p* < 0.0001). The mean BCVAs gained at 100 weeks from the baseline in the VIVID trials was 11.8 letters in the group treated with aflibercept every 4 weeks, and it was 10.6 in the group treated with aflibercept every 8 weeks compared to 5.5 letters in the laser group (*p* < 0.0001 and *p* = 0.0002).

In the VISTA study, 38.3% of the patients received a 2 mg aflibercept every 4 weeks and 33.1% of the patients received the same dosage every 8 weeks; the results showed a gain of ≧15 letters. The visual acuity improved significantly in these groups compared to 13.0% in the laser control group. In the VIVID study, the corresponding changes were 38.2%, 31.1%, and 12.1%, respectively.

A two-year prospective study investigating the therapeutic effects of intravitreal aflibercept, bevacizumab, and ranibizumab found that all three agents improved the visual acuity in eyes with center-involved DME, but the degree of improvement depended on the baseline visual acuity. When the baseline visual acuity was poor (≥20/50), aflibercept was more effective in improving the visual acuity [40].

#### 3.1.3. Bevacizumab

Bevacizumab is used in many patients with DME worldwide because of its efficacy and low cost. Bevacizumab is a recombinant humanized monoclonal immunoglobulin antibody that has two antigen-binding domains. It inhibits VEGF-A. It was originally designed as an anti-tumor agent, e.g., colorectal and non-small cell lung cancers, and it is still in use today. Its molecular weight is 149 × 10^3^, and its estimated terminal half-life is 20 days [42].

The Protocol T randomized clinical trial was conducted by the Diabetic Retinopathy Clinical Research (DRCR) Retina Network to compare the efficacy of bevacizumab, ranibizumab, and aflibercept for DME. A total of 660 patients were randomized in a 1:1:1 fashion to an intravitreal injection every 4 weeks for the first 6 months. Thereafter, the treatments followed a predefined retreatment regimen. Focal macular laser could be performed at 24 weeks for a persistent DME [7,40]. The visual acuity at the 2-year visit improved from the baseline on average by 12.8 letters with aflibercept, 10.0 letters with bevacizumab, and 12.3 letters with ranibizumab. At the 2-year follow-up, the average improvement in visual acuity from baseline was 12.8 letters for aflibercept, 10.0 letters for bevacizumab, and 12.3 letters for ranibizumab (in pairwise comparisons: *p* = 0.02 for aflibercept vs. bevacizumab, *p* = 0.47 for aflibercept vs. ranibizumab, and *p* = 0.11 for ranibizumab vs. bevacizumab). The study concluded that in eyes with better BCVA at the baseline, no difference was detected in the outcomes at the 2-year visit. For the eyes with poorer BCVA at the baseline, the advantage of aflibercept over bevacizumab for the mean BCVA gain persisted through 2 years, although the difference at 2 years was reduced.

Bevacizumab may be slightly less effective than aflibercept for DME but may be selected more often because of its lower cost [43].

#### 3.1.4. Faricimab

Faricimab is an anti-VEGF/anti-angiopoietin-2 (Ang-2) humanized bispecific monoclonal antibody with a molecular weight of 149 × 10^3^. This agent is characterized by simultaneous inhibition of VEGF-A and Ang-2 [44]. The Ang and tyrosine kinase with immunoglobulin-like and epidermal growth factor homology domains (Tie) signalling pathways are the key pathways determining their vascular stability in the retina. Ang-1 (Angiopoietin-1) and Ang-2 are growth factors that play key roles in vessel homeostasis, angiogenesis, and vascular permeability. Under physiological conditions, Ang-1 mediates endothelial cell survival and cell junction integrity through the Tie2 receptors. In retinal vascular diseases, upregulation of Ang-2 competitively inhibits Ang-1 binding to Tie2, thereby neutralizing the vasoprotective effects of the Ang-1 and Tie2 signalling pathways [45]. Ang-1 is a Tie2 receptor agonist, and Ang-1/Tie2 signalling promotes vascular stabilization in healthy vessels. In addition, Ang-2 competes with Ang-1 for Tie2 and induces endothelial cell destabilization [46].

The binding affinities of faricimab to VEGF-A165 and VEGF-A121 are comparable to that of ranibizumab, but its binding affinity is less than that of aflibercept [44]. However, faricimab is a monoclonal antibody against both VEGF-A and Ang-2 that are important factors that affect the permeability and stability of the retinal vessels. Thus, faricimab is expected to have therapeutic effects that are different from other anti-VEGF agents.

The efficacy and safety of faricimab for DME were evaluated in the YOSEMITE and the RHINE Studies [45]. These studies were randomized, double-masked, non-inferiority trials across 353 sites worldwide. A total of 1891 patients were enrolled in the YOSEMITE (n = 940) and the RHINE (n = 951) studies. These trials had a 96-week treatment period, and a final study visit at week 100. The primary endpoint was the mean change in the best-corrected visual acuity (BCVA) at 1 year. There were three groups in these trials. The group treated with faricimab every 8 weeks initially received intravitreal faricimab injections every 4 weeks until week 20 (a total of six injections), followed by a fixed dosing schedule of faricimab every 8 weeks until week 96. The group with personalized faricimab treatment interval (PTI) initially received intravitreal faricimab injections every 4 weeks until week 12 (a total four injections), followed by an adjustable dosing schedule of faricimab every 16 weeks until week 96. The group treated with aflibercept every 8 weeks initially received intravitreal aflibercept injections every 4 weeks until week 16 (a total of five injections), followed by a fixed dosing schedule of faricimab every 8 weeks until week 96. The adjusted mean changes in the BCVA from the baseline (Early Treatment Diabetic Retinopathy Study; ETDRS letters) were 10.7, 11.6, and 10.9, respectively, in the YOSEMITE study, and 11.8, 10.8, and 10.3 in the RHINE study. The results of these trials showed that faricimab was not inferior to aflibercept. In both the YOSEMITE and RHINE studies, the non-inferiority of the PTI groups was achieved with fewer interval-determining visits and extended dosing, with more than 50% of patients receiving faricimab every 16 weeks at week 52 and with more than 60% at 96 weeks [45,47].

One of the characteristics of the faricimab molecule that contributes to its clinical safety is that its Fc region was engineered to abolish its bindings to all Fc gamma receptors (FcγR) and neonatal Fc receptors (FcRn) [44]. The binding with FcγR resulted in complement-dependent cytotoxicity (CDC), antibody-dependent cytotoxicity (ADCC), and antibody-dependent cell phagocytosis (ADCP) [48]. It was expected to reduce the risk of intraocular inflammation by the elimination of the binding sites to FcγR. FcRn is the neonatal Fc receptor for IgG, and it protects IgG in the lysosomes from degrading. This explains its long half-life in the serum [49]. Faricimab does not have a binding site with FcRn and is considered to have a short serum half-life due to the lack of IgG recycling. Faricimab has been shown to be cleared from the systemic circulation faster than the wild-type IgG1 that can bind to FcRn [44]. This rapid systemic clearance of anti-VEGF agents may reduce the systemic adverse effects such as cardiovascular events.

#### 3.1.5. Brolucizumab

Brolucizumab is a humanized anti-VEGF monoclonal antibody with a single-chain Fv fragment (scFv) that inhibits all isoforms of VEGF-A binding to the VEGF receptors, VEGF-1 and VEGF-2. Its molecular weight is 26 × 10^3^ which is much lower than that of ranibizumab, and it has been suggested to have greater tissue penetration. Brolucizumab had a 2.2-fold higher exposure in the retina and a 1.7-fold higher exposure in the retinal pigment epithelium (RPE)/choroid than ranibizumab in rabbits [34]. The time to the maximum concentration in the retina was 1 to 6 h for brolucizumab [34] compared with 6 h for ranibizumab in monkeys [50] and 24 h for aflibercept in rabbits [51]. In addition, the characteristics of this agent are its low molecular weight and high clinical effective dosage (6.0 mg) compared to other anti-VEGF agents. The higher molecular weight and lower effective dosage (0.5 mg to 2.0 mg) compared to the other anti-VEGF agents results in 11 to 22 times more molecules per injection [34].

The efficacy and safety of brolucizumab for DME compared to aflibercept were evaluated in the KESTREL and the KITE studies [52]. These trials were double-masked, multicenter, active-controlled, randomized trials. The primary endpoint was the BCVA change from the baseline to that at 52 weeks.

In the brolucizumab treatment group, the study eyes underwent a loading phase consisting of five doses administered every 6 weeks (at Weeks 0, 6, 12, 18, and 24). Afterward, the eyes received intravitreal injections every 12 weeks (q12w), with the option to adjust the dosing frequency to every 8 weeks (q8w) for the remainder of the study if disease activity was detected during predefined assessment visits (such as a loss of ≥5 letters in BCVA accompanied by an increase in central subfield thickness compared to the subject’s disease status at Week 28). In the aflibercept treatment group, the eyes received five monthly loading doses (at Weeks 0, 4, 8, 12 and 16), followed by a fixed dosing schedule of every 8 weeks (q8w). The least squares mean estimate of visual acuity improvement was +9.2 letters in the brolucizumab 6 mg arm compared with +10.5 letters in the aflibercept arm. The difference was −1.3 letters (95% CI: [−2.9, 0.3]) after adjustments for the baseline BCVA categories and age categories in the KESTREL study. The number of injections at Week 52 in the brolucizuma 6 mg arm and the aflibercept 2 mg arm were 6.8 and 8.5, respectively, in the KESTREL study, and 7 and 8.5 in the KITE study. These results showed that brorucizumab had a non-inferior effect to aflibercept despite the reduced number of injections.

Serious intraocular side effects, e.g., intraocular inflammation (IOI) and retinal vasculitis, have been reported following the use of brolucizumab. In the HAWK and HARRIER studies, clinical trials of brosucizumab for age-related macular degeneration (AMD), the incidence of definite/probable intraocular inflammation (IOI) was 4.6%, that of IOI + vasculitis was 3.3%, and that of IOI + vasculitis + occlusion was 2.1%. There were eight cases (0.74%) of a moderate visual acuity reduction (≥15 ETDRS letters) in eyes with IOI and seven in eyes with IOI + vasculitis + occlusion. On the other hand, the incidence of IOI in the aflibercept-treated eyes was 1.1% with a moderate visual acuity reduction in 0.14% [53]. These findings are obtained from the clinical trials for AMD, but the results from clinical trials for DME appear to be slightly different. The incidence of ocular serious adverse events including IOI, vasculitis, and occlusion was 3.7% for 3 mg brolucizumab, 1.1% for 6 mg brolucizumab, and 2.1% for aflibercept in the KESTREL study. The comparable findings in the KITE study were 2.2% for 6 mg brolucizumab and 1.7% for aflibercept. Although there were no significant differences in the incidence of serious ocular adverse events between brolucizumab and aflibercept in the clinical trials for DME, caution is still required [52].

### 3.2. Topical Corticosteroid Treatment

Topical corticosteroid treatments are known to enhance the progression of cataracts leading to a lack of improvement in the BCVA. However, steroid treatments have been found to be as effective as anti-VEGF treatments when limited to cases of pseudophakic eyes [54].

Dexamethasone (DEX), fluocinolone acetonide (FA), and triamcinolone acetonide (TA) have been used as topical steroid treatments for DME, and their effectiveness has been reported. DEX is used as a sustained release implant and for intravitreal injections, FA is utilized as a sustained release implant, and TA is used for intravitreal and subtenon injections. All have been reported to be effective [55,56,57,58,59] and safe [60]. All steroids have been shown to improve the BCVA and the macular edema to some extent, but the incidence of adverse events such as cataract progression and IOP elevation were also significantly higher than in the control sham laser treatments and anti-VEGF treatment groups [55,56,57,61,62,63,64,65,66,67,68,69,70,71,72,73]. However, there has not been a study on a direct comparison between steroids for these adverse events. A meta-analysis of 10 trials that evaluated the efficacy and safety of intravitreal steroid treatments showed that cataracts progressed in about 2 of 10 participants in the control group compared to 5 to 6 of the 10 participants treated with steroids (moderate certainty evidence at 12 to 36 months). An increase in the IOP was found in 1 of 20 participants in the control group and about 3 of 10 participants in the steroid-treated group (moderate certainty evidence at 12 to 36 months) [74]. However, the need for glaucoma surgery remained rare in patients treated with steroids.

In another meta-analysis comparing the effects of DEX implant, FA implant, and TA injections on DME showed that all types of intravitreal corticosteroids were effective for treating DME, and higher dosages (TA ≥ 4 mg, FA implant of 0.5 µg/day, and DEX implant of 700 µg) had greater improvements in the BCVA [75]. The results showed that TA injections were not inferior to FA or DEX implants for improving the BCVA or decreasing the CMT in DME patients.

Among the three types of intravitreal corticosteroid treatments, the European Society of Retina Specialists’ (EURETINA) study recommended that DEX be used first, and FA may be appropriate for nonsteroid responders with chronic macular edema who were not responsive to the other treatments. Because TA was not approved at the time of the study and caused an increase in the IOP and cataracts, it should be used only in patients who cannot obtain the approved agents for this indication.

The efficacies of a DEX implant, FA implant, and TA injection for DME are good according to the 2017 American Diabetes Association (ADA) Position Statement [76] and the EURETINA guidelines [77]. However, these treatments are rarely used as first-line therapy in eyes with central-involved DME (CIDME) because of the poorer visual acuities as well as the increased adverse events of cataract and glaucoma associated with steroid use.

In Japan, serial injections of anti-VEGF agents are the first-line therapy, although DEX and FA are not approved, and treatments with TA also play a secondary role [78,79]. In the EURETINA guidelines, topical corticosteroid therapy is recommended for nonresponders who have been treated with three to six injections of some anti-VEGF agents. Topical corticosteroids have also been used in patients who have a history of major cardiovascular events.

A single subtenon injection of TA as a pretreatment for PRP can prevent PRP-induced foveal thickening and visual dysfunction in patients with severe diabetic retinopathy and good vision [80]. In Japan, retinal experts claim that subtenon injections of triamcinolone (STTA) may also be helpful in preventing the PRP-induced macular thickening and visual disturbances in the PDR associated with DME [79].

It should be noted that there are some steroid preparations that cannot be used or are not used very often, depending on the country or region. TA injections are not available in some countries and regions. In Japan, steroid implants, DEX and FA are not approved and could not be used.

#### 3.2.1. Intravitreal Injection of Triamcinolone Acetonide (IVTA)

The ETDRS showed that treatments by argon laser photocoagulation were beneficial in reducing the risk of vision reductions in cases with clinically significant DME [81]. The photocoagulation reported was performed by a laser as a conventional macula laser treatment for macular edema. An intravitreal injection of triamcinolone acetonide (IVTA) has been reported to be effective in cases refractory to the conventional macular laser treatment [82]. A prospective, randomized, double-masked clinical trial has shown that IVTA is effective and a relatively safe treatment for DME eyes in the short term [57].

There are several randomized control studies comparing the safety and efficacy of different doses of IVTA in treating clinically significant DME [83,84,85,86,87]. Many studies have concluded that the eyes injected with a larger dose of TA had a greater chance of improving the central retinal thickness (CRT) and BCVA. Clinical studies using high doses of IVTA (≥8 mg) have concluded that the higher doses of IVTA may prolong the duration of the visual benefits in eyes with clinically significant DME more efficiently than lower doses [83,85]. Many of these studies reported that the elevation of the IOP was independent of the IVTA dose. A meta-analysis also reported similar findings [75].

#### 3.2.2. Subtenon Triamcinolone Acetonide Injection (STTA)

IVTA has been reported to be effective in resolving DME. However, intravitreal injections carry considerable risks including acute infectious endophthalmitis and pseudoendophthalmitis. TA delivered by the posterior subtenon route, i.e., STTA, has also been used to treat macular edema due to the Irvine–Gass syndrome and also uveitis. Based on earlier findings, STTA was effective for DME but IVTA significantly improved the visual acuity and the CRT within 3 months. The benefits of either treatment were not significantly different at 6 months, and patients had to be retreated [88].

Although STTA was also associated with the risk of IOP elevations [89], there were significant differences in the frequency of IOP > 30 mmHg in the STTA-treated patients. In the IVTA group, more patients needed antiglaucoma medications than in the STTA group [90].

#### 3.2.3. Intravitreal Sustained-Release Steroid Implants

DEX and FA can also be used as intravitreal sustained-release steroid implants. There are many reports showing the effectiveness of sustained-release steroids. It has been reported that higher doses of FA (0.5 µg/day) and DEX (DEX implant 0.7 mg) were more effective in resolving the DME than lower doses [56,91,92]. The mean change from the baseline in BCVA was 7.5, 6.9, and 5.7 letters at months 3, 6, and 12, respectively, after administration of a 0.5 µg/day of an FA insert, and it was 5.1, 2.7, and 1.3 letters at months 3, 6, and 12, respectively, after the administration of a 0.2 µg/day of FA insert [91]. The average reduction in the CRT from the baseline was greater with a DEX implant of 0.7 mg (−111.6 mm) and a DEX implant 0.35 mg (−107.9 mm) than sham (−41.9 mm; *p* < 0.001) [56].

#### 3.2.4. Switching to Steroid Treatments from Anti-VEGF Treatments

Although steroid treatments are as effective as anti-VEGF treatments, they cannot be first-line treatments because they cannot escape the risk of an elevation of the IOP and exacerbation of cataracts. Therefore, there are no first-line treatments other than anti-VEGF agents. However, the mechanism of action of steroids on DME differs from that of anti-VEGF therapy, and it has been reported that steroid treatment is effective in cases refractory to anti-VEGF treatments.

It has been reported that switching to IVTA from anti-VEGF treatments significantly improved the visual acuity and reduced the central retinal thickness in patients with DME refractory to anti-VEGF therapy [93]. In patients with less than a 10% reduction in the CRT after at least three loading doses of anti-VEGF injections, switching to IVTA or STTA significantly improved both the CRT and the BCVA. In cases where anti-VEGF treatments resulted in a CRT ≤ 300 µm or decreased it to >200 µm but macular edema recurred at least three times at the same interval, switching to TA injections significantly increased the mean time for recurrences from 9.2 ± 2.7 weeks to 22.3 ± 12.9 weeks. The mean interval for a recurrence was extended to more than 8 weeks in 7 of 11 eyes [94].

A multicenter retrospective study reported that the visual acuity and anatomical outcomes at 12 months were significantly better in the group that switched to a DEX implant than the group that continued treatment with anti-VEGF therapy for eyes with DME considered refractory to anti-VEGF therapy after three monthly injections [95].

#### 3.2.5. STTA or IVTA during Cataract Surgery for Eyes with DME

Individuals with diabetes mellitus have been reported to be at a greater risk of developing cataracts than nondiabetic individuals [96]. It has been reported that several cytokines including interleukin-6 (IL-6), IL-8, IL-10, IL-1β, interferon-induced protein-10 (IP-10), monocyte chemotactic protein-1 (MCP-1), and VEGF were significantly elevated in patients with DME following cataract surgery [97]. However, the anti-VEGF agents failed to sustain a significant reduction in the central retinal thickness after cataract surgery [98]. On the other hand, IVTA led to a sustained and significant reductions in the CRT after cataract surgery [98]. This is probably because cytokines other than VEGF were altered at the onset and the progression of the DME after cataract surgery [97]. In addition, the levels of these other cytokines in eyes with DME cannot be effectively decreased by the use of anti-VEGF agents [99]. It is possible to maintain the reduction in the central retinal thickness similarly by performing cataract surgery combined with STTA or IVTA in patients with DME [100]. Although there are no significant differences, the postoperative BCVA and CRT tended to be better with the IVTA combined therapy [100].

### 3.3. Laser Photocoagulation

#### 3.3.1. Focal/Grid Laser

Focal (direct) grid laser photocoagulation has been the standard treatment for DME since its efficacy was shown in 1985 [101]. However, serious complications such as choroidal neovascularization [102], enlargement of the laser scars [103], and subretinal fibrosis [104] have been reported after grid laser treatments. It has been shown that laser photocoagulation destroyed the photoreceptors and had serious risks of central vision reduction. Grid laser treatments have been widely used with special attention paid to these serious complications. However, the frequency of their use has decreased because large-scale clinical trials have shown that anti-VEGF treatment is superior to laser treatments [5,54].

Focal (direct) laser treatment involves the direct application of the laser spot to the sites of the focal fluorecein leakage which are mainly microaneurysms (MAs) [81,101]. This laser treatment protocol carries the risk of producing scotomas in addition to the complications described for other laser lesions and should be performed at some distance from the fovea. There are reports that the presence of MAs is related to cases refractory to anti-VEGF treatments [105,106], but their treatments are important. It is still widely used as an effective laser therapy. This laser treatment is related to the Navigated laser treatment protocol, which will be described below.

#### 3.3.2. Subthreshold Laser Treatments

Focal/grid laser treatments have been shown to be effective for DME and have been widely performed even though they cause retinal scars and other serious complications. However, when exposure times are shorter, such as less than 1 ms, the damage to the retinal pigment epithelium (RPE), neural retina, and choriocapillaris is reduced [107]. A non-damaging laser treatment called subthreshold laser treatment is currently used instead of the grid laser; this treatment improves DME.

There are two methods to apply scan laser (SL) treatments: a short pulse of a continuous wave laser with a pattern-scan laser (PASCAL) device and an endpoint management (EPM) algorithm [108]. The second method is a subthreshold micropulse laser that delivers pulses whose duration is in microseconds [107,109]. The use of these subthreshold laser treatments limits the spread of the heat to the adjacent retinal and choriodal layers, whereas conventional lasers cause greater thermal damage and scaring. It has been reported that both types of treatments are effective in resolving DME without causing any visible changes to the retina. It is believed that these treatments act by stimulating the RPE directly [108,110]. It has been reported that subthreshold micropulse diode laser photocoagulations induced the expression of heat shock protein 70 [111]. The thermal stimulation may be involved in the reconstitution of the RPE which contributes to the reduction in the DME.

##### PASCAL and Endpoint Management (EpM) Algorithm

A PASCAL algorithm capable of applying multiple laser spots in a short period of time was developed by Topcon (Topcon Medical Laser Systems, Santa Clara, CA, USA). EpM is a software with an algorithm to calculate the energy required for a subthreshold laser burn. PASCAL with the EpM algorithm is a subthreshold laser burn with a continuous wave laser in milliseconds [108]. In randomized clinical trials, both the PASCAL subthreshold laser and threshold laser algorithm have been shown to improve the visual acuity and reduce the CRT in eyes with DME [112]. Hamada et al. have shown that the CRT was significantly reduced after a subthreshold laser treatment using PASCAL with the EpM, but there were no significant changes in the macular sensitivity. There were no changes in the autofluorescein images, and none of the eyes had symptoms of scotoma after the photocoagulation [113]. Karasu et al. reported that this treatment protocol was effective for DME refractory to anti-VEGF treatment [114].

##### Micropulse Diode Laser

Micropulse diode laser is a pulse pattern in which the on and off modes are repeated in microseconds. Before beginning, the laser is directed to an area of the retina away from the macula to determine the threshold power at which a burn is barely visible. Then, the laser is set to the determined power and applied. The results of several non-randomized trials [115,116,117,118,119] and prospective, randomized trials [120,121,122,123] reported that this type of micropulse laser application is effective for DME. In a randomized double-masked clinical trial, Lois et al. showed that subthreshold micropulse laser for DME was equivalent to that of a standard continuous wave laser which produced a visible burn on the retina [123].

#### 3.3.3. Navigated Laser

A navigated laser system called Navilas has been developed; it has an eye-tracking laser delivery system. This system enables accurate laser irradiation even for small MAs [124]. Neubauer et al. reported that Navilas decreased the retreatment rate for DME in a study with a short follow-up period [125]. Kato et al. reported that Navilas was effective for the treatment of refractory DME lasting for over 6 months in a retrospective trial of 25 eyes [126]. Nozaki et al. used Navilas^®^ to perform navigated focal laser photocoagulation on the MAs identified by indocyanine green angiography. the authors reported that there were significant improvements of the CRT and the BCVA at 6 months after the treatment [127].

#### 3.3.4. Selective Retina Therapy

Roider et al. reported that multiple short argon laser pulses can coagulate the retinal pigment epithelium (RPE) selectively while sparing the adjacent neural retina and choroid [128]. Brinkmann et al. showed that the origin of the RPE damage for a single-pulse irradiation of pulse durations of up to three microseconds was the damage of the microbubbles around the melanosomes causing a rupture of the cellular structures [129].

It is generally believed that the mechanism for the improvement of the macular edema is the activation of the drainage function of the RPE by the reconstructed monolayer structure through healing of the RPE cells [130]. Selective retinal therapy for DME has been reported to significantly improve the BCVA and the CRT after 6 months [130,131].

### 3.4. Combination Therapy with Laser Photocoagulation and Anti-VEGF Agents

Laser photocoagulation treatments for DME are effective, but their effects are not any better than those of anti-VEGF treatments. This was found in a large clinical trial that compared the effectiveness of anti-VEGF treatments to conventional grid laser treatments such as the RESTORE, the REVEAL, the VISTA, and the VIVID studies [6,132,133]. Therefore, a number of studies have been conducted comparing anti-VEGF monotherapy and combination therapy to laser photocoagulation to assess the effects of add-on laser treatments.

#### 3.4.1. Grid Laser + Anti-VEGF Combined Treatments

The RESTORE [132] and REVEAL [6] studies were large clinical trials of ranibizumab, and the results showed that there were no significant differences in the mean changes in the baseline BCVA and the CRT between ranibizumab monotherapy and ranibizumab + laser combination therapy.

#### 3.4.2. Focal Laser + Anti-VEGF Treatments

Hirano et al. reported on the results of combination therapy using anti-VEGF agents and short pulse focal/grid laser photocoagulation in a prospective multicenter single-arm clinical trial [105]. In this study, 1 week after each intravitreal ranibuzumab (IVR), a short-pulse focal laser to MAs or grid laser was delivered to the thickened retinal areas with capillary nonperfusion or diffuse leakage within the vascular arcades to treat the residual leakage outside of the fovea (>500 μm) and reduce edema fluid influx. Both the BCVA and CRT improvements were comparable between the MA(−) and MA(+) groups. The MA(−) group required significantly fewer pro re nata (PRN) injections than the MA(+) group did over the 6-month study period (mean 3.4 ± 1.6 vs. 5.3 ± 0.9, *p* = 0.023).

#### 3.4.3. Combined Navigated Laser + Anti-VEGF

A randomized clinical trial comparing ranibizumab monotherapy to ranibizumab combined therapy with Navilas reported that the combination therapy was able to reduce the number of ranibuzumab injections significantly [134]. Payne et al. performed a randomized clinical trial comparing monthly treatments with a treat and extend regimen using ranibizumab with and without angiography-guided laser (Navilas) for center-involved DME, the TREX-DME study. The authors reported that adding angiography-guided laser photocoagulation to this dosing regimen did not significantly improve the outcomes at 1 year. The number of injections was 13.1 (treated monthly), 10.7 (treat and extend without Navilas), and 10.1 (treat and extend with Navilas) [135]. At month 12 with this protocol, 29% (without Navilas) and 38% (with Navilas) of the eyes were able to extend the treatment intervals to a maximum of 12 weeks.

#### 3.4.4. Subthreshold Laser + Anti-VEGF

There have been several studies that compared anti-VEGF monotherapy to combined therapy of subthreshold laser and anti-VEGF agents for DME. In all of these studies, the anti-VEGF monotherapy and combination therapy of anti-VEGF + SL were compared, and there were no significant differences observed in the changes in the BCVA and the CRT between the two groups. The combination therapy group had a significantly lower number of anti-VEGF injections in three retrospective studies [136,137,138] and three prospective studies [139,140,141]. However, two studies recently reported that the number of injections of anti-VEGF agents did not differ significantly between the two groups, and that the addition of subthreshold laser did not offer additive effects in reducing the treatment burden or improving the diabetic macular edema [142,143]. These studies had some differences in the treatment regimen, agents, baseline data, and retreatment criteria, which may have affected the results (Table 2). The efficacy of combining subthreshold lasers with anti-VEGF treatment has not been definitively determined.

#### 3.4.5. Targeted Retinal Photocoagulation (TRP) + Anti-VEGF

Although it was not a laser treatment around the macula, the application of VEGF from non-perfused areas (NPAs) is enhanced. Thus, TRP for NPAs guided by fluorecein angiography is expected to prevent the development of DMEs. Clinical trials comparing anti-VEGF monotherapy and TRP + anti-VEGF combined therapy have been conducted. Takamura et al. conducted a 6-month randomized control trial of 52 eyes (bevacizumab vs. TRP + bevacizumab) and found that the combination therapy with TRP and bevacizumab for NPAs was more effective in maintaining a reduced CRT after focal/grid laser photocoagulation for patients with DME [144]. On the other hand, Brown et al. concluded after a 3-year randomized DAVE trial of 40 eyes (ranibizumab vs. TRP + ranibizumab) that there was no evidence that a combination therapy with ranibizumab and TRP improved the visual outcomes and reduced the treatment burden compared with ranibizumab alone [145].

### 3.5. Pars Plana Vitrectomy

Pars plana vitrectomy was an effective and widely performed procedure to treat DME before the anti-VEGF agents were introduced. The efficacy of vitrectomy for DME was first reported by Lewis et al. They reported that the visual acuity improved in 9 of the 10 eyes with thickened and taut premacular posterior hyaloid membrane after pars plana vitrectomy with separation of the posterior hyaloid. It was suggested that the release of macular traction by vitrectomy was the reason for the improvement in the DME [146]. Vitreous surgery was considered suitable for eyes with diabetic macular traction and the edema associated with a thickened and taut posterior hyaloid. La Heij et al. studied 21 eyes with DME without evidence of macular traction from a thickened vitreous membrane. They performed vitrectomy with the creation of a posterior vitreous detachment and found a resolution of the DME with an improvement of the visual acuity in the majority of cases [147]. In eyes without a posterior vitreous detachment, there was vitreous traction even without a thickened vitreous membrane which may have been involved in the exacerbation of the macular edema. Kumagai et al. reported on the efficacy of vitrectomy combined with the creation of a posterior vitreous detachment for the treatment of DME without a thickened and taut posterior hyaloid in a clinical trial of 496 eyes for up to 5 years [148]. They concluded that pars plana vitrectomy with or without internal limiting membrane (ILM) peeling was beneficial in eyes with diffuse nontractional DME. Gandorfer et al. reported on the results of pars plana vitrectomy with peeling of the ILM in 12 eyes with diffuse DME. They reported that the visual acuity improved by at least two lines in 11 eyes [149]. They concluded that vitrectomy with the removal of the ILM leads to an accelerated resolution of a diffuse DME with an improvement of the visual acuity without subsequent epiretinal membrane formation. In evaluating the effects of vitrectomy on DME using data collected in a prospective clinical study conducted by the DRCR network, greater visual acuity improvement occurred in eyes after the removal of an epiretinal membrane (ERM) (*p* = 0.006), and a greater reduction in the CRT occurred in eyes with the removal of the ILM (*p* = 0.003). However, the visual acuity improvement in eyes with removal of the ILM was not significantly better (*p* = 0.15) [150].

#### 3.5.1. Efficacy of Vitrectomy for DME Refractory to Anti-VEGF Treatment

Vitrectomy was performed only in a limited number of cases once the efficacy of anti-VEGF therapy was recognized, and a consensus was reached that anti-VEGF should be the first-line therapy. There is a report that anti-VEGF treatments can improve the visual acuity more than vitrectomy even in cases with thickened posterior vitreous membrane and vitreomacular traction [151]. Thus, it appears reasonable to initially perform anti-VEGF therapy unless the patients have markedly thickened membranes considering the risk of complications.

On the other hand, the effectiveness of vitrectomy for DME refractory to anti-VEGF therapy has been reported [152,153,154]. It is believed that vitrectomy may be a suitable treatment option for DME patients who are refractory to anti-VEGF therapy or had multiple recurrences after the anti-VEGF therapy [79].

Mukai et al. reviewed the medical records of 27 eyes with DME that had undergone vitrectomy, and they reported that vitrectomy was effective in eyes that had limited response to anti-VEGF agents and STTA prior to the vitrectomy. They also stated that it is likely more effective for DME accompanied by an epiretinal membrane or vitreomacular traction [153]. Vikas et al. performed a prospective study evaluating the outcomes of pars plana vitrectomy with ILM peeling on eyes with DME unresponsive to anti-VEGF therapy and IVTA. They reported that vitrectomy resulted in good anatomical outcomes and the results were comparable in eyes with DME with and without a tractional component [154]. Hwang et al. reported on the three-year outcomes of vitrectomy combined with intraoperative dexamethasone implantation for non-tractional refractory DME. They reported that vitrectomy combined with intraoperative dexamethasone implantation led to satisfactory long-term clinical outcomes and the number of intraocular injections was reduced [152].

#### 3.5.2. New Surgical Procedures

There are reports of new surgical procedures that are effective in cases with cystoid macular edema (CME) that are refractory to anti-VEGF treatment. Tachi et al. reported on the efficacy of cystotomy for diabetic CME [155]. Singh et al. and Asahina et al. also reported on the efficacy of cystoid puncture or cystotomy for refractory diabetic CME [156,157]. Imai et al. found that a fibrinogen-rich component existed in the cystoid lesions of some patients, and en bloc removal of these lesions was effective in the treatment of CME [158]. Morizane et al. reported on a surgical procedure that was effective for diffuse DME but not for CME [159]. In this procedure, a small retinal detachment was made in the macula by injecting a balanced salt solution (BSS) into the subretinal space with a 38-gauge needle after a vitrectomy. Before completing the surgery, fluid–air exchange was performed. The authors reported that this planned foveal detachment technique facilitated a rapid resolution of the macular edema and contributed to the improved visual acuity. There have also been reports on the efficacy of subretinal injections of ranibizumab [160], and this method was expected to increase the therapeutic effect by using ranibizumab which has a therapeutic effect against DME. The authors reported that the CRT decreased from 498.58 ± 152.16 µm preoperatively to 365.74 ± 120.12 µm at 6 months by this surgical procedure. Kumagai et al. reported that the removal of foveal hard exudate by subretinal injections of balanced salt solution with a 38-gauge needle during vitrectomy was effective [161]. The foveal hard exudates decreased in all 7 cases soon after the surgery. The visual acuity improved by >0.2 logMAR units in six eyes and was unchanged in one eye.

### 3.6. Agents for Systemic Treatments

There are several agents that have been used systemically, viz., oral medications, that have been shown to be effective against DME. While there are some therapeutic agents that improved the DME, there are others that have been reported to exacerbate the DME.

The use of pioglitazone [162,163,164] and insulin [165] preparations has been reported to increase the risk of exacerbation of the macular edema. It has been reported that there was a lack of a significant longitudinal association between thiazolidinediones, including pioglitazone, and the incidence and progression of diabetic eye diseases in a 4-year follow-up study of 2856 participants in the ACCORD trial [166]. In addition, semaglutide has been reported to have a risk of enhancing the progression of retinopathy, although not macular edema [167]. In addition, the HIF-PH inhibitors have been cited as being significantly associated with adverse events such as retinal hemorrhages and macular edema [168].

On the other hand, the drugs that improve DME include sodium glucose cotransporter-2 (SGLT2) inhibitors and fenofibrates. These agents improve the DME with a relatively high level of evidence.

#### 3.6.1. SGLT2 Inhibitor

Almost all of the glucose entering the glomeruli in the afferent glomerular arterioles is filtered into the nephron fluid of the proximal renal tubules. Up to 90% of this filtered glucose is reabsorbed by SGLT2 in the initial proximal convoluted segment (S1) of the proximal tubule. The remaining glucose is reabsorbed from the filtrate in the more distal convoluted and straight segments by sodium glucose cotransporter-1 (SGLT1) [169]. SGLT2 inhibitors (SGLT2i) promote urinary excretion of glucose by inhibiting glucose reabsorption by SGLT2. Not only does SGLT2i have therapeutic effects on diabetes by lowering blood glucose, but it also has been shown to have protective effects on the heart [170] and kidneys [171]. Recently, it was reported to have a protective effect on the retina [172,173,174]. There are case reports [175,176,177], a cohort study [178], and a retrospective study [179] documenting that SGLT2i treatments improve the DME. However, the mechanism of SGLT2i affecting DME and diabetic retinopathy has not been determined conclusively. SGLT2i has hypoglycemic and diuretic effects, but these effects alone are not thought to affect diabetic retinopathy and DME.

Chung et. al. reported that the results of their cohort study showed that the hazard ratio (95% CI) for diabetic retinopathy was 0.89 (0.83–0.97) for SGLT2i initiators; it was compared to dipeptidyl peptidase-4 inhibitor (DPP4i) initiators. They also showed that SGLT2is might be associated with a lower risk of diabetic retinopathy compared with DPP4i [180]. Su et. al. suggested that the use of SGLT2i was associated with a lower risk of DME in Type 2 DM compared to the use of glucagon-like peptide-1 receptor agonists (GLP-1RA) [178]. The lower risk of occurrence of diabetic retinopathy and DME with SGLT2i cannot be explained by the diuretic effect alone. It was suggested that there are other mechanisms of action of SGLT2i.

SGLT2 has been shown to exist not only in the proximal renal tubules but also in the bovine retinal pericytes [181]. Thus, the pericytes may act as functional glucose sensors in the retinal microvascular circulation [182]. Hanaguri et al. showed that SGLT2i (tofogliflozin) improved the response of glial activation and VEGF expression by immunofluorescence. They also showed that the implicit times of the oscillatory potentials of the ERGs were significantly shortened in SGLT2i-treated db/db mice [172]. Hu et al. reported that the protein of SGLT2 was expressed on mice and human retinal microvascular endothelial cells, and SGLT2i (dapagliflozin) reduced the apoptosis of diabetic mice retina independently of hypoglycemia [173]. Matthews et al. showed that SGLT2i reduced the retinal abnormalities associated with diabetic retinopathy, vascular leakage indicated by lower albumin staining and reduced expression of the pathogenic factor VEGF in the retina in model mice [174]. These findings indicated that the improvement of diabetic retinopathy and DME was brought about by not only the blood glucose improvement and diuretic effects of SGLT2i, but also the possibility that SGLT2i acts directly on the retina of patients with diabetes mellitus.

#### 3.6.2. Fenofibrate

Fenofibrate is a peroxisome proliferator-activated receptor alpha (PPARα) agonist. This agent is used to treat hyperlipidemia by lowering triglyceride levels and increasing the levels of high-density lipoprotein cholesterol [183]. Fenofibrate can reduce the free fatty acid levels by upregulating the synthesis of molecules for fatty acid transport and oxidation through the activation of PPARα [184].

Randomized clinical trials evaluating the association between diabetic retinopathy and fenofibrate include the Fenofibrate Intervention and Event Lowering in Diabetes (FIELD) study and the Action to Control Cardiovascular Risk in Diabetes (ACCORD) study. The ACCORD study reported that intense glycemia was controlled by fenofibrate, but it did not enhance the blood pressure control and reduce the progression of retinopathy [185]. The findings from the FIELD study demonstrated that the fenofibrate-treated group had a significantly reduced risk (HR 0.66, 95% CI 0.47–0.94; *p* = 0.022) of experiencing the composite endpoints, which included a two-step progression of retinopathy grade, development of macular edema, or the need for one or more laser treatments (in either eye) compared to the placebo group [186]. In both studies, fenofibrate reduced the risk of the progression of diabetic retinopathy. Fenofibrate also reduced the risk of DME, although its potential effectiveness was not determined.

#### 3.6.3. Metformin

Metformin is the main agent for antidiabetic treatment, and it reduces the level of glucose without causing hypoglycemia [187]. Metformin may have the potential to induce the splicing of VEGF-A mRNA into the VEGF120 isoform leading to reduced activation of VEGFR2 [188], and it may inhibit the translation of VEGF-A protein by inducing the expression of microRNAs that specifically target VEGF-A [189]. Metformin stimulates the adenosine monophosphate-activated protein kinase pathway to protect the photoreceptors and the retinal pigment epithelium. This protection was associated with a decrease in the oxidative stress, decrease in the DNA damage, and increase in the mitochondrial energy production [190].

Uwimana et al. studied 109 patients retrospectively and reported that a combination of metformin and anti-VEGF agents decreased the risk of anti-VEGF resistance in DME patients. They defined anti-VEGF resistance as a persistent macular edema with a decrease in the CRT to ≤25% after three anti-VEGF injections. The mean CRT of the non-metformin group decreased from 344.88 ± 129.48 µm to 318.29 ± 123.23 µm (20.85%) and from 415.64 ± 144.26 µm to 277.11 ± 99.25 µm (31.51%) (*p* = 0.031) in the metformin group. Moreover, the metformin group had fewer patients resistant than the non-metformin, 24 (45.3%) versus 41 (73.2%) [191].

### 3.7. Recommended Treatment for DME

In this section, the recommended treatments of DME are proposed after consideration of the clinical evidence reviewed in Section 3.1, Section 3.2, Section 3.3, Section 3.4, Section 3.5 and Section 3.6. The pathology of DME is complicated, and there are various therapeutic approaches, and new drugs and treatment methods have been developed. the number of treatment options is still increasing. Each patient with DME has a different medical history; patients range from those who are treatment-naïve to those that have undergone different types of medications and surgical treatments.

Anti-VEGF treatment is generally the first-line treatment for center-involved DME. Laser photocoagulation such as photocoagulaton of MAs is recommended for non-center-involved DME. Anti-VEGF therapy should be considered first for treatment naïve center-involved DME. There are currently no definitive criteria for the selection of anti-VEGF agents: ranibizumab, aflibercept, bevacizumab, faricimab, brolucizumab, and other anti-VEGF agents. Decisions should be made in consideration of the risk of systemic and local ocular complications, and the economic burden of the treatments must be considered.

The anti-VEGF treatment is initiated with 3–5 loading doses every 4 weeks (every 6 weeks for brolucizumab), followed by PRN or a treat and extend (TAE) or a fixed intervals regimen.

If the systemic risk such as cardiovascular events and financial burden cannot be tolerated, topical corticosteroid treatment is the first candidate. There is a risk of cataract progression and intraocular pressure elevation with this treatment, and the patients must be carefully monitored.

Next to be considered is laser photocoagulation and vitrectomy. When MAs are the major cause of the DME, photocoagulation of the MAs is a very useful treatment. If there are no MAs, a minimally invasive method such as a subthreshold laser is recommended. Pars plana vitrectomy is an effective treatment in cases with thickened and taut premacular posterior hyaloid membrane and vitreomacular traction seen in the OCT images.

For DME refractory to one anti-VEGF agent, switching to other anti-VEGF agents such as brolucizumab and faricimab should be considered. Furthermore, combination therapy with corticosteroids, laser photocoagulation, and pars plana vitrectomy should be considered depending on the pathology and failed prior treatments.

There are limited countries and regions where agents for systemic treatment are available for DME treatment, and their active use is not yet recommended. However, if there are indications for the use of these agents for systemic conditions, it is considered effective to perform treatment in cooperation with medical institutions that are performing the systemic treatments.

## 4. Conclusions

DME is associated with retinopathy, vasculitis, and neuropathy. Many studies have reported that the neurological changes appear first in diabetic retinopathy, and the macular edema, the retinal angiopathy, edema, and neuropathy are closely related to each other. It is important to provide neuroprotection by resolving the edema, and the treatments for the edema may improve the neurological impairments. The pathology of DME is complex, and various therapeutic approaches, new drugs, and treatment methods have been developed; in addition, the number of treatment options is increasing. The treatment of DME, which is a type of neurosis, is still being refined, and the establishment of the optimal treatment methods has still not been completed. Currently, it is necessary to combine multiple treatments according to the pathology of each individual case.

## Figures and Tables

**Figure 1 ijms-24-09591-f001:**
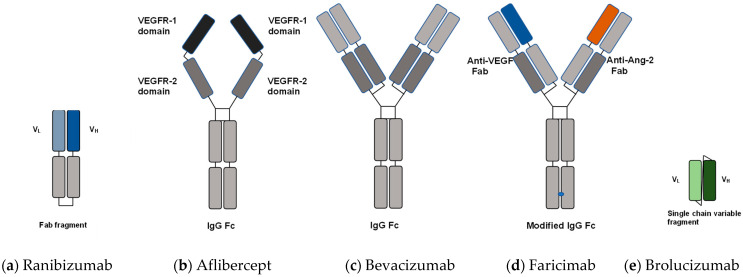
Molecular structure of anti-VEGF agents [33,34,35]. (**a**) Ranibizumab is the Fab fragment of a humanized monoclonal antibody against VEGF. (**b**) Aflibercept is a recombinant protein composed of Fc fragment of human IgG, human VEGFR-1 and VEGFR-2. (**c**) Bevacizumab is a recombinant humanized monoclonal IgG antibody that has two antigen-binding domains. (**d**) Faricimab is an anti-VEGF/anti-Ang-2 humanized bispecific monoclonal antibody. (**e**) Brolucizumab is a humanized anti-VEGF monoclonal antibody with a single-chain F_V_ fragment that inhibits all isoforms of VEGF-A binding to the VEGF receptors. V_L_: light chain variable domain, V_H_: heavy chain variable domain, Fab: fragment antigen binding, VEGFR: vascular endothelial growth factor receptor, IgG: immunoglobulin G, Fc: fragment crystallizable, VEGF: vascular endothelial growth factor, Ang-2: angiopoietin-2.

**Table 1 ijms-24-09591-t001:** Comparisons of molecular characteristics and general clinical information for anti-VEGF agents [33,34,35].

Anti-VEGF Agents	Ranibizumab	Aflibercept	Bevacizumab	Faricimab	Brolucizumab
Molecular format	Fab fragment	VEGFR1/2-Fc fusion protein	Full antibody (lgG1)	anti-VEGF/anti-Ang-2 humanized bispecific monoclonal antibody	Single-chain variable fragment
Molecular weight	~48 × 10^3^	97–115 × 10^3^	~149 × 10^3^	150 × 10^3^	26 × 10^3^
Binding molecule	VEGF-A	VEGF-A, VEGF-B, PlGF-1, PlGF-2	VEGF-A	VEGF-A, Ang-2	VEGF-A
Clinical dose	0.50 mg	2.0 mg	1.25 mg	6.0 mg	6.0 mg
Relative number of molecules per injection	0.5–0.6	1.0	0.4–0.5	1.15	11.2–13.3

VEGF: vascular endothelial growth factor, Fab: fragment antigen binding, VEGFR: vascular endothelial growth factor receptor, Fc: fragment crystallizable, IgG: immunoglobulin G, VEGF: vascular endothelial growth factor, Ang-2: angiopoietin-2, PlGF: placental growth factor.

**Table 2 ijms-24-09591-t002:** Earlier studies comparing combination therapy of anti-VEGF and subthreshold laser/anti-VEGF monotherapy for DME.

Author/Journal/Year	Material and Protocol Study Period	Retreatment Criteria	Study Design Follow-Up Periods	Patient Inclusion Criteria of CRT and BCVA	CRT (µm) Baseline ⇒ Final * Difference	BCVA (logMAR) Baseline ⇒ Final * Difference	No. of Injections	† *p*
Moisseiev E et al. Eur J Ophthalmol 2018 [136]	IVR + SML vs. IVR Jan 2013–Jun 2015		Retrospective 12 months	n = 38 (19 vs. matched control 19) All patients had no more than 3 prior IVR	IVR 408.4 ⇒ 335.9 72.5	0.41 ⇒ 0.39 0.02	5.6/12 months	<0.001
					IVR + SML 316.8 ⇒ 282.6 34.2	0.29 ⇒ 0.24 0.05	1.7/12 months	
Altınel MG, et al. Lasers Med Sci. 2021 [137]	IVB + SML vs. IVB Sep 2017–Mar 2020		Retrospective 15 months	n = 80 (40 vs. 40) Excluded intravitreal injections within the preceding 6 months, CRT > 250 µm	IVB 384.7 ⇒ 325.8 58.9	0.39 ⇒ 0.32 0.07	8.65/15 months	<0.05
					IVB + SML 379.2 ⇒ 292.6 86.6	0.38 ⇒ 0.25 0.13	7.38/15 months	
El Matri L, et al. Ther Adv Ophthalmol. 2021 [138]	IVB + SML vs. IVB 3 + PRN Jan 2015–Jan 2019	BCVA ⩽ 20/25 Presence of IRF and/or SRF	Retrospective 12 months	n = 98 eyes (49 vs. 49) (63 patients) Treatment naïve for DME CRT ⩽ 500 µm, BCVA ⩾ 20/400	IVB 359.9 ⇒ 305.9 54.0	0.60 ⇒ 0.49 0.11	7.2/12 months	<0.005
					IVB + SML 479.1 ⇒ 289.6 189.5	0.69 ⇒ 0.50 0.19	4.1/12 months	
Khattab AM et al. Graefes Arch Clin Exp Ophthalmol 2019 [139]	IVA + SML vs. IVA 3 + PRN Feb 2017–Dec 2018	CRT > 250 µm	Prospective 18 months	n = 54 eyes (27 vs. 27) (51 patients) Excluded intravitreal injections within the preceding 6 months, CRT > 250 µm, BCVA: 20/400–20/40	IVA 462.0 ⇒ 249.5 212.5	‡ 31.7 ⇒ 50.6 18.9 (0.378)	7.3/18 months	<0.005
					IVA + SML 457.1 ⇒ 244.6 212.5	‡ 35.0 ⇒ 54.8 19.8 (0.396)	4.1/18 months	
Kanar HS et al. Ind J Ophthalmol 2020 [141]	IVA + SML vs. IVA 3 + PRN Apr 2015–Nov 2017	20% increase in CRT 1-line decrease at BCVA	Prospective 12 months	n = 56 (28 vs. 28) Treatment naïve for DME CRT ≧ 300 µm, BCVA: 0.2–0.9	IVA 451.28 ⇒ 328.8 122.5	0.38 ⇒ 0.20 0.18	5.39/12 months	<0.001
					IVA + SML 466.07 ⇒ 312.0 154.1	0.40 ⇒ 0.17 0.23	3.21/12 months	
Abouhussein MA et al. Int Ophthalmol 2020 [140]	IVA + SML vs. IVA 3 + PRN period: not stated	CRT ≧ 300 µm	Prospective 15 months	n = 40 (20 vs. 20) Treatment naïve for DME CRT ≧ 300 µm, BCVA > 3/60	IVA 457.9 ⇒ 290.5 167.4	0.70 ⇒ 0.24 0.46	8.4/15 months	0.029
					IVA + SML 469.6 ⇒ 288.5 181.1	0.76 ⇒ 0.20 0.56	7.5/15 months	
Koushan K et al. Clin Ophthalmol 2022 [142]	IVA + SML vs. IVA 1 (Continue until ME resolves) + PRN Mar 2017–Oct 2018	10% change in CRT 1-Snellen-line change at BCVA	Prospective 12 months	n = 30 (15 vs. 15) Excluded intravitreal injections within the preceding 120 days, CRT > 310 µm, BCVA: 20/400–20/30	IVA 433.4 ⇒ 288.3 145.1	0.38 ⇒ 0.32 0.06	8.5/12 months	0.61
					IVA + SL 457.8 ⇒ 289.5 168.3	0.36 ⇒ 0.22 0.14	7.9/12 months	
Tatsumi T et al. Sci Rep 2022 [143]	IVA + SL vs. IVA 3 + PRN Sep 2016–Sep 2020	100 µm increase in CRT 2-line decrease at BCVA	Prospective 24 months	n = 51 (25 vs. 26) Excluded intravitreal injections within the preceding 90 days, CRT > 300 µm, BCVA: 0.05–0.7	IVA 442.8 ⇒ 319.5 123.3	0.37 ⇒ 0.32 0.05	5.86/24 months	0.86
					IVA + SL 472.8 ⇒ 329.5 143.3	0.48 ⇒ 0.28 0.20	6.05/24 months	

*: This value is the difference between the mean baseline value and the mean final value, not the mean change from baseline to final. †: *p*-value for statistical analysis of the number of anti-VEGF injections for monotherapy and combination therapy with subthreshold laser. ‡: BCVAs are described in ETDRS letters in this column. Values of change in BCVA in parentheses are equivalent to BCVA (logMAR). CRT: central retinal thickness, BCVA: best corrected visual acuity, LogMAR: logarithm of the minimum angle of resolution, anti-VEGF: anti-vascular endothelial growth factor, SML: subthreshold micropulse laser, IVR: intravitreal injection of ranibizumab, IVB: intravitreal injection of bevacizumab, IVA: intravitreal injection of aflibercept, 3 + PRN: initial 3 monthly injections and pro re nata, IRF: intraretinal fluid, SRF: subretinal fluid, DME: diabetic macular edema, ETDRS: Early Treatment Diabetic Retinopathy Study.

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
