# Peer review of "Current Treatments for Diabetic Macular Edema"

_ijms, 2023, doi:10.3390/ijms24119591_

Round 1

Reviewer 1 Report

This review article has discussed the pathology and currently existing treatment options for diabetic macular edema (DME) with a precise emphasis on diabetic retinopathy (DR). The review provides adequate details on the topic. However, the author is requested to address the following comments.

1.     In Table 1, please leave a space between the numbers and units. For instance, 1.25mg should be written as 1.25 mg. This comment is applicable to rows about molecular weight and clinical dose as well.

2.     Please give a brief explanation about the anti-VEGF agents in Figure 1 legend.

3.     On page 6, line 247, a space is required between anti-VEGF and agents.

4.     On page 7, line 278, this sentence (occlusion was 3.7% for 3.8 mg brolucizumab mg,..) is perplexing. Please make it clear.

5.     On page 8, line 350, please correct the spelling for ‘clinicaly’.

6.     On page 8, line 371, please delete ‘of FA’ from ‘0.5 μg/day of FA’.

7.     On page 10, line 424, please correct ‘haves been’.

8.     Please add reference(s) for section 3.4.1. (Page 11).

9.     On page 12, line 553, a space is required before (Table 2).

10.  On page 17, line 668, please change ‘has been’ to ‘have been’.

11.  On page 17, line 671, please add ‘of’ before ‘diabetic eye diseases’.

12.  On page 18, line 742, ‘Metfolmin’ should be corrected as ‘Metformin’.

13.  Please spell out the abbreviations at their first mention. Some of the abbreviations have not been spelled out in a few places in the text.

14. Overall, a critical check for typing/grammatical errors is needed throughout the text.

Minor editing of English language required

Reviewer 2 Report

This is an excellent review of all of the literature, data, and scientific rationale for the pathophysiology and all of our existing treatments for diabetic retinopathy. This will an excellent resource for those looking for a concise review and for trainees. Topical steroids are not widely used in the US, but their inclusion as a potential treatment is interesting and appropriately referenced. Perhaps some attention could be given to the lack of use of brolucizumab in the US and lack of availability of several of the treatment modalities mentioned (triesence, kenalog) and also to the black box warning against the use of intraocular kenalog. The mention of new laser therapies and also combination therapy is important and well done.
